



**Contrasting ambient fine particles hygroscopicity derived by HTDMA and HR-AMS measurements**
**between summer and winter in urban Beijing**
Xinxin Fan[1], Fang Zhang[1*], Lu Chen[1], Don Collins[2], Weiqi Xu[3, 4], Xiaoai Jin[1], Jingye Ren[1],Yuying Wang[1, 5],
Hao Wu[1], Shangze Li[1], Yele Sun[3, 4], Zhanqing Li[1, 6]
*[1]State Key Laboratory of Earth Surface Processes and Resource Ecology, College of Global Change and*
*Earth System Science, Beijing Normal University, Beijing 100875, China*
*[2]Department of Chemical and Environmental Engineering, University of California Riverside, Riverside,*
*California, USA*
*[3]State Key Laboratory of Atmospheric Boundary Layer Physics and Atmospheric Chemistry, Institute of*
*Atmospheric Physics, Chinese Academy of Sciences, Beijing 100029, China*
*[4] College of Earth Sciences, University of Chinese Academy of Sciences, Beijing 100049, China*
*[5]School of Atmospheric Physics, Nanjing University of Information Science and Technology, Nanjing*
*210044, China*
*[6]Earth System Science Interdisciplinary Center and Department of Atmospheric and Oceanic Science,*
*University of Maryland, College Park, Maryland, USA*
Correspondence to: Fang.zhang@bnu.edu.cn
**Abstract**
The effects of aerosols on visibility through scattering and absorption of light and on climate through
altering cloud droplet concentration are closely associated with their hygroscopic properties. Here, based on
field campaigns in winter and summer in Beijing, we compare the size-resolved hygroscopic parameter ($\kappa_{gf}$)
of ambient fine particles derived by an HTDMA (Hygroscopic Tandem Differential Mobility Analyzer) to
that (denoted as $\kappa_{chem}$) of calculated by an HR-ToF-AMS (High-resolution Time-of-Flight Aerosol Mass
Spectrometer) measurements using a simple rule with a uniform internal mixing hypothesis. We mainly
focus on contrasting the disparity of $\kappa_{gf}$ and $\kappa_{chem}$ between summer and winter to reveal the impact of
atmospheric processes/sources on aerosols hygroscopicity and to evaluate the uncertainty in estimating
particles hygroscopicity with the hypothesis. We show that, in summer, the $\kappa_{chem}$ for 110, 150 and 200 nm
particles was averagely ~10% - 12% lower than $\kappa_{gf}$, with the greatest difference between the values observed
around noontime when aerosols experience rapid photochemical aging. In winter, no apparent disparity
between $\kappa_{chem}$ and $\kappa_{gf}$ is observed for those >100 nm particles around noontime, but the $\kappa_{chem}$ is much higher
than $\kappa_{gf}$ in the late afternoon when ambient aerosols are greatly influenced by local traffic and cooking





sources. By comparing with the observation from other two sites (Xingtai, Hebei and Xinzhou, Shanxi) of
north China, we verify that atmospheric photochemical aging of aerosols enhances their hygroscopicity and
may induce a coating effect which thereby leads to 10%-20% underestimation of the hygroscopic parameter
if using the uniform internal mixing assumption. The coating effect is found more significant for these >100
nm particles observed in remote or clean regions. However, local primary sources, which result in an
externally mixture of the fine particles with a large number of POA (Primary Organic Aerosol) in urban
Beijing, makes the particle much less hygroscopic and cause 20-40% overestimation of the hygroscopic
parameter by the mixing rule assumption. In addition, we also note lower $\kappa_{chem}$ than $\kappa_{gf}$ for 80, 110 and 150
nm particles during the nighttime of winter, particularly in polluted days, probably due to a nighttime
coating effect driven by condensation of secondary hygroscopic species on pre-existing aerosols in cold
season. Our results suggest that it is critical to parameterize the impacts in model simulations to improve the
evaluation of the aerosols indirect effect.
**1.  Introduction**

The effects of aerosols on visibility through scattering and absorption of light and on climate through

altering cloud droplet concentration are influenced by their hygroscopic growth. Understanding and
reducing the uncertainty in prediction of the aerosol hygroscopic parameter ($\kappa$) using chemical composition
would improve model predictions of aerosol effects on clouds and climate.

The hygroscopic parameter, $\kappa$, is dependent upon particle chemical composition (Gunthe et al., 2009).

The hygroscopic properties of an aerosol, in addition to being affected by its chemical composition, are also
affected by the particle mixing state. The mixing state of aerosol particles can be divided into external
mixing and internal mixing. The chemical components in the aerosol particles are independent of each other.
The chemical composition of the different types of aerosol particles is different within a certain particle size
range, and the mixed state is external mixing. As the aerosol particles undergo transport, coagulation, and
aging/coating in the atmosphere, the chemical components become more uniformly mixed within each
particle size range, with the aerosol mixing state approaching an internal mixture. Depending on the physical
properties of the different aerosol components, internal mixing can be divided into uniform internal mixing





and "core-shell" mixing (Jacobson, 2001). For uniform internal mixing the distribution of the chemical
components is the same throughout each particle. "Core-shell" mixing refers to a mixing state in which
certain chemical components are coated or coagulated on the surface of other chemical components (such as
black carbon) during aging. In the atmosphere, the aerosol sources and sinks are varied, the physical and
chemical processes experienced by the aerosols are complex, and the mixing state is more complicated. In
heavily polluted areas, BC is usually mixed with other chemical components. Freshly emitted BC is mostly
in an external mixed state. With the aging process, it gradually transforms into the internal mixing state
(Chen, et al., 2016; Lee, et al., 2015; Wang, et al., 2017). Based on observations in the winter of Beijing
urban area, Wang et al. (2019) found that the secondary aerosol generated from photochemical reactions is
thickly coated on the surface of BC.

Studies have shown that the difference between the $\kappa$ obtained using H-TDMA data, $\kappa_{gf}$, and that

calculated based on the volume mixing ratio of chemical components, $\kappa_{chem}$, depends on the mixing state and
the extent of aging of the particles (Mikhailov, et al., 2015; Zhang et al., 2017). Results from Cruz and
Pandis (2000) also indicate that $\kappa_{gf}$ of internally mixed ammonium sulfate and organic matter is higher than
$\kappa_{chem}$ calculated for assumed uniform internal mixing. Similarly, in some studies on aged aerosols
(Bougiatioti, et al., 2009; Chang, et al., 2007; Kuwata, et al., 2008; Wang, et al., 2010), the concentration of
CCN was underestimated by the calculation based on uniform internal mixing. Our previous study
demonstrated that particle mixing state has large impacts on prediction of CCN concentration for the aerosol
sampled in Beijing (Ren et al., 2018). Zhang et al. (2017) observed an evident underestimation of around
noontime particle hygroscopicity based on chemical composition measurements in urban Beijing. Wang et al.
(2018a) also noted a lower hygroscopic parameter estimated by the simple volume mixing ratio than that
derived from direct HTDMA measurement at a site in North China Plain. These studies have revealed the
uncertainty in the estimation of aerosol hygroscopicity parameters using chemical composition volume
mixing ratios with the assumption that of uniform internal mixing, but there is still lack of a systematic
investigation on the cause and magnitude of the effect. Furthermore, most studies that have been conducted
compare the size-resolved hygroscopic parameter $\kappa$ obtained with an HTDMA with $\kappa$ calculated from bulk



chemical composition measurements, motivating our analysis that employs size-resolved chemical
composition measured by an HR-ToF-AMS.

The aim of this paper is to study the hygroscopicity and mixing state characteristics of fine particles in

the Beijing urban area, and to reveal the impact of atmospheric processes/sources on aerosols hygroscopicity
and elucidate the uncertainty in calculating the hygroscopic parameter using simple mixing rule estimates
based on size-resolved chemical composition. The experiment and theory in the study are introduced in Sect.
2. The comparison between the hygroscopic parameter obtained from the HTDMA and and that calculated
using size-resolved chemical composition is discussed in Sect. 3. Conclusions from the study are given in
Sect. 4.

## 2. Experiment and Theory

### 2.1. Site and instruments

In this study, we mainly focus on analysis of the data obtained from two campaigns in urban Beijing

(BJ: 39.97 ° N, 116.37 ° E). In addition, we also compare the results from the field campaigns with those
from two sites, Xingtai (XT: 37.18 °N,114.37 °E), and Xinzhou (XZ: 38.24 °N,112.43 °E), in North China
Plain (Fig. 1). The BJ site is located at the Institute of Atmospheric Physics (IAP), Chinese Academy of
Sciences, which is between the north third and fourth ring roads in northern Beijing. Local traffic and
cooking emissions can be important at the site (Sun et al., 2015). The sampling period in cold season was
from 16 November to 10 December 2016, during the domestic heating period in Beijing. The sampling
period in warm season was from 25 May to 18 June 2017. The XT site is located in the National
Meteorological Basic Station, which is about 17 km from the XT urban area. The sampling period was from
17 May to 14 June 2016. Xingtai, with a high level of industrialization and urbanization, is located in the
center of the North China Plain. Due to industrial emissions and typically weak ventilating winds,
concentrations of $PM_{2.5,}$ black carbon and gaseous precursors are extremely high at the Xingtai site (Fu et al.,
2014). Xinzhou is located north of Taiyuan and about 360 km southwest of Beijing, in the north central part
of Shanxi Province, and is surrounded by mountains on three sides. The XZ site is located in a town,





surrounded by agricultural land (such as corn fields). Local emissions from motor vehicles and industrial
activities have relatively little influence on the sampled aerosol (Zhang et al., 2016). Because of its location
and elevation,   the aerosol at the XZ site is usually aged and transported from other areas. The sampling
period was from July 22 to August 26, 2014 at XZ site.

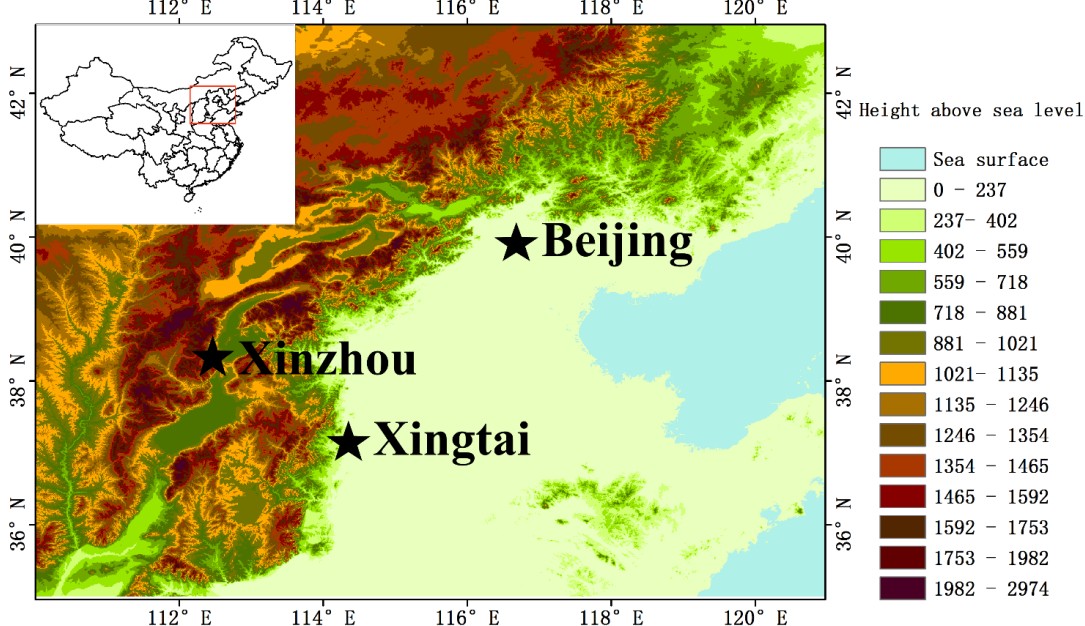


Figure 1. The map location of the sites


Particle number size distribution (PNSD) in the size range from 10 nm to 550 nm was measured with a

Scanning Mobility Particle Sizer (SMPS; Wang & Flagan, 1990; Collins et al., 2002), which consists of a
long differential mobility analyzer (DMA, model 3081L, TSI Inc) to classify the particle and a condensation
particle counter (CPC, model 3772, TSI Inc.) to detect the size classified particles. The sampled particles
were dried to relative humidity < 30% before entering the DMA. The measurement time for each size
distribution was five minutes.

The HTDMA system used in this study has been described in detail in previous publications (Tan et al.,

2013; Wang et al., 2017; Zhang et al., 2017). Here, only a brief description is given. A Nafion dryer dried



125 the sampled particles to relative humidity < 20%, after which the steady state charge distribution was

126 reached in a bipolar neutralizer. The first differential mobility analyzer (DMA$_1$, model 3081L, TSI Inc.)

127 selected the quasi-monodisperse particles through applying a fixed voltage. The dry diameters selected in

128 this study were 40, 80, 110, 150, and 200 nm. The quasi-monodisperse particles were humidified to a

129 controlled RH (90% in this study) using a Nafion humidifier. A second DMA (DMA$_2$, same model as the

130 DMA$_1$) coupled with a water-based condensation particle counter (WCPC, model 3787, TSI Inc.) measured

131 the particle number size distributions of the humidified aerosol. RH calibration with ammonium sulfate was

132 carried out regularly during the study.

133   The hygroscopic growth factor (Gf) is defined as the ratio of the mobility diameter at a given RH to the

134 dry diameter:

$$\text{Gf} = \frac{D(RH)}{D(dry)}$$

135   The Gf probability density function is retrieved based on the TDMA$_{inv}$ algorithm developed by Gysel et

136 al. (2009). Dry scans in which the RH between the two DMAs was not increased were used to define the

137 width of the transfer function.

138   Size-resolved non-refractory submicron aerosol composition was measured with an Aerodyne

139 high-resolution time-of-flight aerosol mass spectrometer (HR-ToF-AMS; Xu et al., 2015). The particle

140 mobility diameter was estimated by dividing the vacuum aerodynamic diameter from the AMS

141 measurements by particle density. Because the uncertainty caused by the fixed density across the size range

142 is negligible (Wang et al. 2016), here, the particle density is assumed to be 1600 kg m$^{-3}$ (Hu et al., 2012).

143 AMS positive matrix factorization (PMF) with the PMF2.exe (v4.2) method was performed to identify

144 various factors of organic aerosols. Xu et al. (2015) have described the operation and calibration of the

145 HR-ToF-AMS in detail. Black carbon (BC) mass concentration was derived from measurements of light

146 absorption with a 7-wavelength aethalometer (AE33, Magee Scientific Corp.; Zhao et al., 2017).





**2.2. Data**
The time series of the submicron particle mass concentration $PM_1$, (Fig. 2a), bulk mass concentrations
of the main species in $PM_1$ (Fig. 2b), mass fraction of the chemical composition of $PM_1$ (Fig. 2c), and
Gf-PDFs for 40, 80, 110, 150, 200 nm particles (Fig. 2d-h) during the campaign are presented in Fig. 2. As
shown in Fig. 2, quite distinct temporal variability of aerosol chemical and physical properties was observed
between winter and summer. The average mass concentration of $PM_1$ was 55.2 $\mu g/m^3$ in the winter and 16.5
$\mu g/m^3$ in the summer during our study periods. In this study, we define the conditions when the mass
concentration in winter period was $< 20 \ \mu g \ m^{-3}$ and $>80 \ \mu g \ m^{-3}$ as clean and polluted conditions, respectively.
Organic aerosol (OA), consisting of secondary organic aerosol (SOA) and primary organic aerosol (POA),
was the major fraction during both the winter and summer sampling periods. POA concentration was higher
than that of SOA in the winter, which reflects the influence of primary emissions such as coal combustion
OA (COOA) in Beijing (Hu et al., 2016; Sun et al., 2016). In contrast, SOA usually dominated in the
summer, which is evidence that secondary aerosol formation played a key role in the source of $PM_1$. Figs.
2d-h show the time series of the probability density functions (PDFs) of GF for 40, 80, 110, 150, and 200
nm particles, respectively. Distinct hydrophobic (with Gf of ~1.0) and more hygroscopic (with Gf of ~1.5)
modes were observed from Gf-PDFs of both small and large particles. Sometimes the more hygroscopic
mode particles were more concentrated and at others the hydrophobic particles were. In general though, the
more hygroscopic mode dominated for larger particles (i.e. 150 and 200 nm), and the less hygroscopic mode
did for the smallest particles (e.g. 40 nm) (Fig. 2d-h and Fig. S1). Occasionally, only the hydrophobic mode
was evident for 150 and 200 nm particles, which occurred when POA dominated the $PM_1$. Only the
hygroscopic mode was discernable for 40 nm particles during new particle formation (NPF) events that
occurred more frequently in summer than winter (Fig. S2).





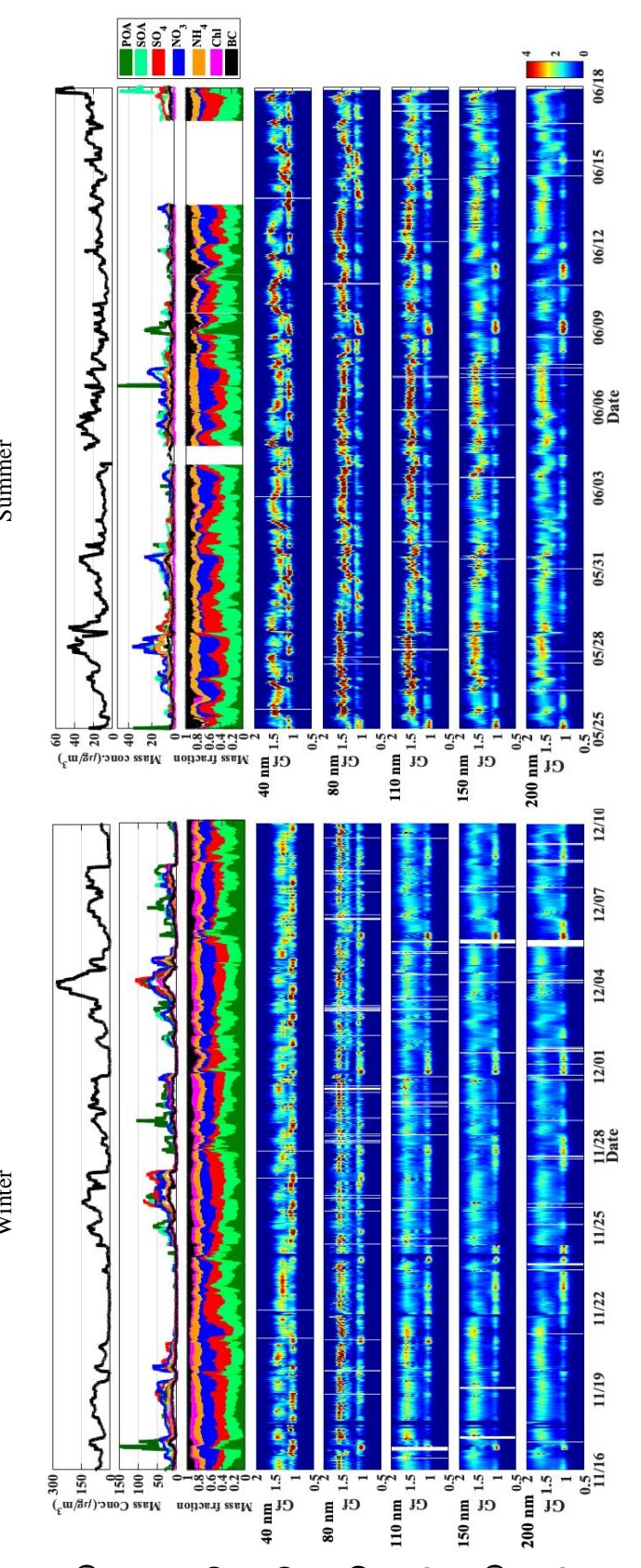

Figure 2. Winter (left) and summer (right) time series of (a) mass concentration of PM1; (b) bulk mass concentration of the main species in PM1; (c)

mass fraction of the chemical composition of PM1; (d-h) Gf-PDFs for 40, 80, 110, 150 and 200 nm particles.





**2.3. Theory and method**
2.3.1 Derivation of the hygroscopic parameter, $\kappa$, from the growth factor (Gf)
According to $\kappa$-Köhler Theory (Petters and Kreidenweis, 2007), the hygroscopicity parameter $\kappa$ can be
derived using the growth factor measured by an HTDMA.
$$\kappa = (Gf^3 - 1)\left(\frac{\exp\left(\frac{A}{D_d Gf}\right)}{RH} - 1\right) \ , \tag{1}$$

$$A = \frac{4\sigma_{s/a} M_w}{RT \rho_w} \ , \tag{2}$$

where Gf is hygroscopic growth factor measured by HTDMA, $D_d$ is the dry diameter of the particles, RH is
the relative humidity in the HTDMA (90%, in our study), $\sigma_{s/a}$ is the surface tension of the solution/air
(assumed here to be the surface tension of pure water, $\sigma_{s/a} = 0.0728$ N m$^{-2}$), $M_w$ is the molecular weight of
water, R is the universal gas constant, T is the absolute temperature, and $\rho_w$ is the density of water.
2.3.2 Derivation of the hygroscopic parameter, $\kappa$, from chemical composition data
For an assumed internal mixture, $\kappa$ can also be calculated by a simple mixing rule on the basis of
chemical volume fractions (Petters and Kreidenweis, 2007; Gunthe et al., 2009):
$$\kappa_{chem} = \sum_i \varepsilon_i \kappa_i \ , \tag{3}$$

$$\kappa_{org} = f_{POA} * \kappa_{POA} + f_{SOA} * \kappa_{SOA} , \tag{4}$$

where $\kappa_i$ and $\varepsilon_i$ are the hygroscopicity parameter and volume fraction for the ith individual (dry) component
in the mixture, respectively, and $f_{POA}$ and $f_{SOA}$ are the volume fractions of POA and SOA in the organic
component. The AMS provides mass concentrations of organics and of many inorganic ions. The inorganic
components mainly consisted of $(NH_4)_2SO_4$ and $NH_4NO_3$ (Zhang et al., 2014; Zhang, et al., 2016; Zhang et
al., 2017). As noted above, the organic components mainly consisted of POA and SOA. According to



previous study, the values of $\kappa$ are 0.48 for $(NH_4)_2SO_4$ and 0.58 for $NH_4NO_3$ (Petters and Kreidenweis,
2007). To estimate $\kappa_{org}$ we used the following linear function derived by Mei et al. (2013): $\kappa_{org} = 2.10 \times f_{44}$
– 0.11. We derived the volume fraction of each species by dividing mass concentration by its density. The
values of density are 1720 kg m$^{-3}$ for $(NH_4)_2SO_4$ and 1770 kg m$^{-3}$ for $NH_4NO_3$. The densities of all organics,
POA, and SOA are assumed to be 1200 kg m$^{-3}$ (Turpin et al., 2001), 1000 kg m$^{-3}$ , and 1400 kg m$^{-3}$
respectively. The $\kappa$ and density ofBC are assumed to be 0 and 1700 kg m$^{-3}$. In the following discussions, $\kappa_{gf}$
and $\kappa_{chem}$ denote the values derived from HTDMA measurements and calculated using the ZSR mixing rule,
respectively.
**3. Results and discussion**
**3.1. Diurnal variations of ambient fine particles physiochemical properties and hygroscopic growth**
**factor**

The diurnal variations of the PNSD, mass concentration of PM$_1$, mass concentration and fraction of

chemical components in PM$_1$, and Gf-PDFs for 40 and 150 nm particles during the campaign are shown in
Fig. 3. During the summer an obvious peak value in the PNSD is observed around noontime due to NPF
events that typically started around 10:00 LT (Local Time). The resulting sharp increase in number
concentration of nucleation mode particles was followed by decreased concentration and a rapid growth in
diameter of the particles along with increased mass concentration of SOA and sulfate in PM$_1$, indicating
strong photochemical and secondary formation processes during daytime in the summer. In contrast, NPF
was not evident during the winter period, which may in part be due to the much higher (~3x) PM$_1$   mass
concentrations in the winter than in the summer.   Note that peak values in number concentration and in
mass concentrations of PM$_1$ and POA occurred during the early evening (17:00-21:00, LT) indicating the
strong impact of local sources from traffic emissions and cooking. In addition, the diurnal cycles of aerosol
physical and chemical properties are also influenced by the diurnal changes in the planetary boundary layer
(PBL) that leads to accumulation of particles during nighttime when higher values of both number and mass
concentration were observed.





Fig. 3e shows the diurnal variations of the Gf-PDFs for 40 nm and 150 nm particles. Owing to the
continued local and primary emissions near the study site, the Gf-PDFs for 40 nm particles generally display
a bimodal shape with more and less hygroscopic modes (with Gf of ~ 1.5 and ~ 1.1 respectively) throughout
the day both in winter and summer periods, indicating an external mixing state for the 40 nm particles. Note
that, during nighttime and early morning in the winter, the more hygroscopic mode dominated and was
shifted to higher Gf than during the daytime. This is thought to be due to heterogeneous/aqueous reactions
on pre-existing primary small particles, and/or coagulation/condensation processes that are enhanced at
night under lower ambient temperature and higher relative humidity, all of which result in a more
hygroscopic and more internally-mixed aerosol (Liu et al., 2011; Massling et al., 2005; Ye et al., 2013; Wu
et al., 2016; Wang et al., 2018a).). Interestingly, in the summer period, the concentration of the hydrophilic
mode increased quickly around noontime and in the early afternoon (12:00-16:00), with a corresponding
decrease in the relative concentration of the hydrophobic mode, which likely indicates a transformation of
the particles from externally to internally mixing state as a result of the species condensation from the
photochemical reaction (Wu et al., 2016; Wang et al., 2017), resulting in an increase in particle
hygroscopicity (Fig. S3). For 150 nm particles, the hygroscopic mode in the Gf-PDF is more dominant
during daytime in particular during the summer period when the strong solar radiation promotes
photochemical aging and growth, thus producing a more internally-mixed aerosol. The dominant
hydrophobic mode at around 18:00 was observed both in winter and summer and reflects abundant traffic
emissions and cooking sources (primarily with POA) during the early evening period (Fig. 2c).



Figure 3. Diurnal variations in (a) particle number size distribution; (b) mass concentration of PM$_1$; (c) bulk

mass concentration of main species in PM$_1$; (d) mass fraction of chemical composition of PM$_1$; (e) Gf-PDFs

for 40 and 150 nm particles in winter and summer period respectively.

**3.2 $\kappa_{gf}$ dependence on D$_p$**

The size dependence of particle hygroscopicity parameters for the winter and summer periods are

presented in Fig.4. In the winter, the 40 nm particles were least hygroscopic and the hygroscopicity of larger




particles (>80 nm) displayed insignificant dependence on particle size. The size independence for the larger
particles is consistent with the observed similarity in mass fractions of inorganic and organic species across

**(a)**

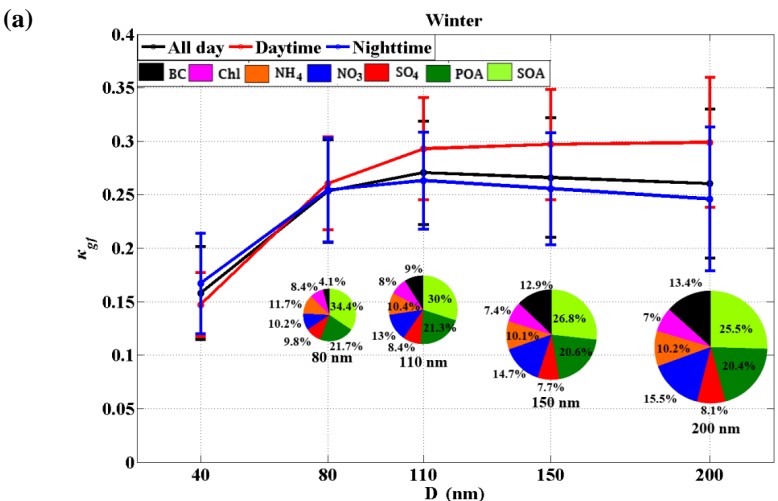


**(b)**

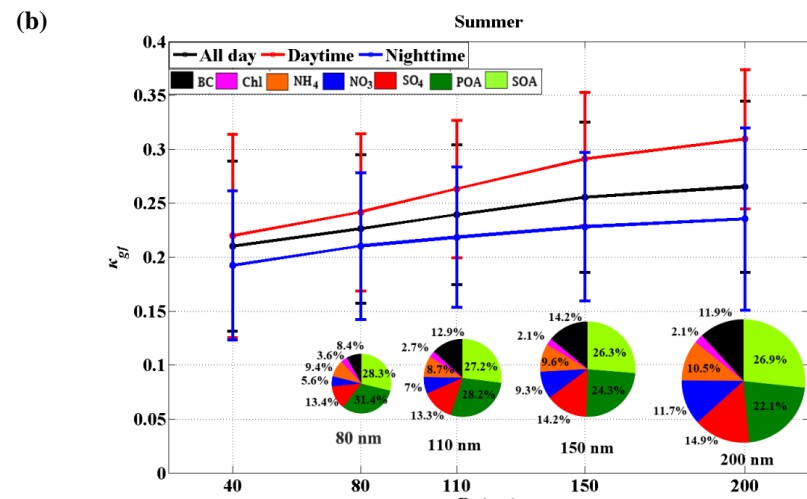


Figure 4. The dependence of $\kappa$ on $D_p$ at the urban Beijing site during the study periods. The $\kappa$ values are
retrieved from the size-resolved HTDMA measurements. The error bars represent ±1σ. The size-resolved
chemical mass fractions at the corresponding $D_p$ is also presented.
the size range as shown in the pie charts in Figure 4a. A similar dependence of particle hygroscopicity on
particle size was also observed in the urban area of Beijing during the wintertime of 2014 (Wang et al.,
2018b). In the summer, hygroscopicity increased with increasing particle size, which is expected based on



the size dependent patterns shown in the pie charts, with the mass fraction of POA decreasing with the
particles size and the mass fraction of inorganics like sulfate and nitrate increasing with particle size.
**3.3. Closure of HTDMA and chemical composition derived $\kappa$**
A closure study was conducted between $\kappa_{chem}$ and $\kappa_{gf}$ (Fig. 5) to investigate the uncertainty of the two
methods, and especially to further illustrate whether particle hygroscopicity can be well predicted by $\kappa_{chem}$
calculated by assuming internal mixing. Since a size-resolved BC mass concentration measurement was not
available during the campaign, we use the bulk mass fraction of BC particles measured by the AE33
combining with size-resolved BC distribution in Beijing reported by Liu et al. (2018) to estimate $\kappa_{chem}$.

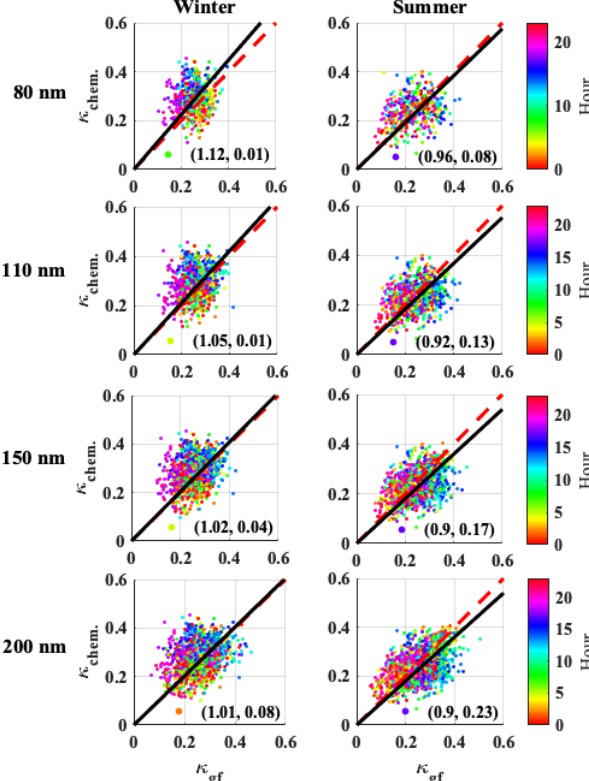


Figure 5. Closure of $\kappa_{chem}$ calculated from size-resolved chemical composition data and $\kappa_{gf}$ retrieved from
hygroscopic growth factor by HTDMA measurements in winter (left panels) and summer (right panels)
period. The dots with different color correspond to observed time of a day during the campaign as shown by
the color bar. The number in parentheses is correlation coefficients ($R^2$) and slopes of linear fits.




Uncertainty in $\kappa$ is due in part to measurement uncertainty of the HTDMA/CCNc system and
uncertainty resulting from non-ideality effects in the solution droplets, surface tension reduction due to
surface active substances, and the presence of slightly soluble substances that dissolve at RH higher than
that maintained in the HTDMA (e.g., Wex et al., 2009; Good et al., 2010; Irwin et al., 2010; Cerully et al.,
2011; Wu et al., 2013).    However, our previous study demonstrated that, for this region, estimates using
HTDMA data are still better than those using the simple mixing rule based on chemical volume fractions for
an assumed internal mixture (Zhang et al., 2017). Therefore, here we focus on discussing and exploring the
uncertainty of $\kappa_{chem}$ by taking $\kappa_{gf}$ as the reference.
Our results show that, in winter, the slopes from linear fitting of $\kappa_{chem}$ and $\kappa_{gf}$ are about 0.96-1.0 for
particles with diameters of 80, 110, 150, and 200 nm, indicating an overall consistency of $\kappa_{chem}$ and $\kappa_{gf.}$ In
summer, the slopes are 0.88-0.89 for 110, 150, and 200 nm particles, meaning there is about 10% - 12%
underestimation of $\kappa_{chem}$. However, the poor correlations (typically with correlation coefficients, $R^2$, of < 0.3)
between $\kappa_{chem}$ and $\kappa_{gf}$ of the 80, 110, 150, 200 nm particles both in winter and summer reflect large
uncertainty in one or both of the calculated parameters. The large uncertainties are likely due to the
unreasonable assumption of particle mixing state (e.g. Cruz and Pandis, 2000; Svenningsson et al., 2006;
Sjogren et al., 2007; Zardini et al., 2008), which varies with their aging and other physiochemical processes
in the atmosphere. For example, underestimation of $\kappa_{chem}$ for the summer occurred mostly in the afternoon.
This may be associated with photochemical processes at around noontime. More specific investigations of
the particle mixing and aging impacts on $\kappa_{chem}$ will be further addressed in the following sections.
**3.4 Atmospheric processes and sources effects indicated by diurnal cycles of $\kappa_{chem}$ and $\kappa_{gf}$**
The diurnal cycles of particle hygroscopicity in the summer and winter with the use of the size-resolved
chemical composition observations and the ratio of $\kappa_{chem}$ to $\kappa_{gf}$ are shown in Fig. 6. In summer, at
09:00-15:00, the disparity between $\kappa_{chem}$ and $\kappa_{gf}$ is insignificant for smaller particles (80 and 110 nm), both
of which show slight decrease from 09:00 or 10:00 to 12:00-13:00 due to the frequent NPF event that
usually corresponds to a large fraction of organics (Fig. 3) in urban Beijing. For larger particles (150 and
200 nm), the disparity between $\kappa_{chem}$ and $\kappa_{gf}$ around noontime and in the early afternoon is very significant,





corresponding to >20% underestimation of particle hygroscopisity by $\kappa_{\mathrm{chem}}$ (with the ratio of $\kappa_{\mathrm{chem}}$ to $\kappa_{\mathrm{gf}}$ of
~0.8). Similar patterns were also noted by Zhang et al., (2017) but which is only based on a comparison
between $\kappa_{\mathrm{chem}}$ derived from bulk chemical composition and $\kappa_{\mathrm{gf}}$. Our results further clarify that the rapid
photochemical aging of BC particles, which are generally with dominant size modes of 100-200 nm in the
atmosphere, leads to the core-shell structure in which certain secondary aerosol generated from
photochemical reactions is thickly coated on the surface of BC (Wang et al., 2019). The hygroscopicity of
the coated BC particles may only depend on the coating layer (Ma et al., 2013), thus resulting in the
noontime/early afternoon underestimation of particle hygroscopicity by $\kappa_{\mathrm{chem}}$. While, no significant
differences between $\kappa_{\mathrm{chem}}$ and $\kappa_{\mathrm{gf}}$ are observed during night time. Note that $\kappa_{chem}$ is slightly higher than $\kappa_{gf}$
during early evening traffic rush hour and cooking time, when emissions of primary hydrophobic particles
(e.g. POA) are high (Fig. 3b), thus resulting in a large percentage of externally-mixed particles (Fig. 3e, Fig.
S4 and Fig. S5). Therefore, the assumption of uniform internal mixing will overestimate hygroscopicity
according to our previous study (Zhang et al., 2017). But the particles experience rapid conversion and
mixing in urban Beijing due to high precursor gases (Sun et al., 2015; Wu et al., 2016; Ren et al., 2018), and
thus the coated/aged particles produced through photochemical processing in the afternoon can mix and
interact with and freshly emitted primary particles emitted during rush hour (Wu et al., 2008). Therefore,
during nighttime (22:00-06:00, LT), the particles are more uniform internally-mixed, which is reflective of
the assumption for calculation of $\kappa_{chem}$, a much better consistency between $\kappa_{chem}$ and $\kappa_{gf}$ is observed. And due
to the relatively clean conditions overall in the summer, no large differences are observed under clean and
polluted conditions (Fig. S5-S7).

Figure 6. Diurnal variations in (a) $\kappa_{chem}$ using size-resolved chemical composition data and $\kappa_{gf}$ in winter and summer period; and (b) ratio of $\kappa_{chem}$ to $\kappa_{gf}$ in winter and summer period.





In winter, the disparity between $\kappa_{chem}$ to $\kappa_{gf}$ is insignificant at 09:00-15:00 due to the weakening

effect of photochemical aging. From 15:00 to 21:00 LT, due to the strong vehicle and cooking sources

around the site, the particles are dominated by the hydrophobic mode with a large concentration of

externally-mixed POA particles (Fig. 3 and Fig. S8), the calculated $\kappa_{chem}$ is much higher than $\kappa_{gf}$, with the

maximum ratio of $\kappa_{chem}$ to $\kappa_{gf}$ of 1.2-1.4, and the greatest disparity is observed for small particles. The

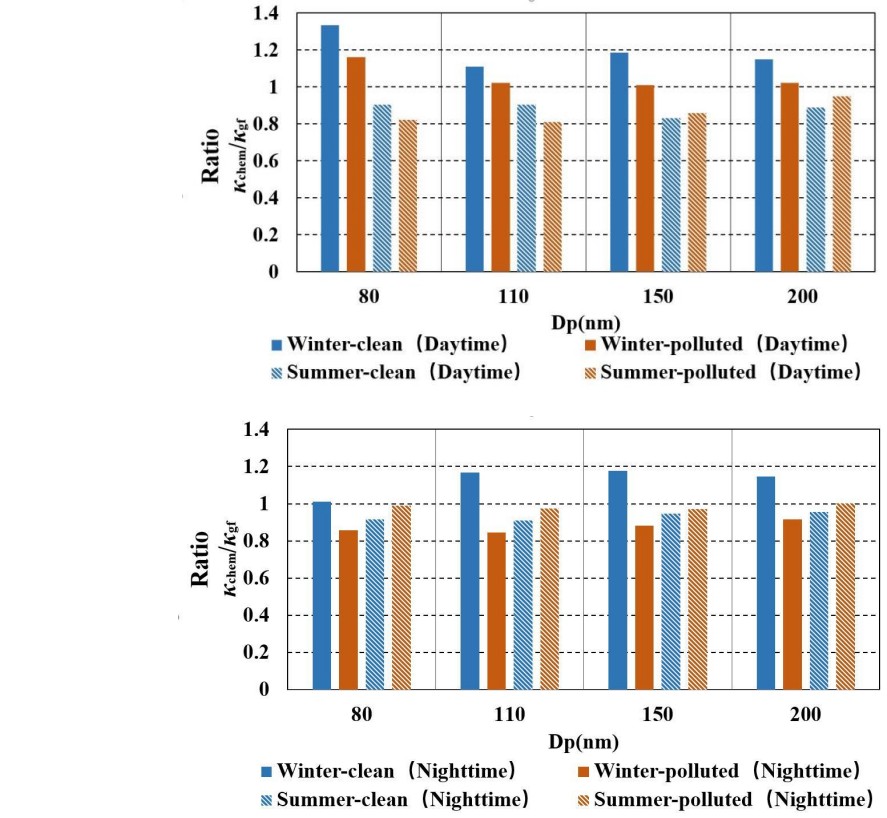

Figure 7. Ratio of mean $\kappa_{chem}$ to $\kappa_{gf}$ during daytime (top panel) and nighttime (bottom panel) under

clean /polluted conditions between winter and summer period.

disparity is further enhanced during clean periods (Fig. S7, Fig. S9 and Fig. 7) when the hydrophobic mode

is dominant (Fig. 8). But note that during the nighttime, $\kappa_{chem}$ is slight lower than $\kappa_{gf}$, with the minimum ratio

of $\kappa_{chem}$ to $\kappa_{gf}$ of ~0.8 for 80 nm particles and ~0.9 for 110 and 150 nm particles at 02:00-04:00 LT (Fig. 6b),

indicating an underestimation of particle hygroscopicity using composition data. The disparity at nighttime

is further increased during heavily polluted events (Fig. 7 and Fig. S9), when the particles are more


internally-mixed with only one hygroscopic mode (Fig. 8 and Fig. S8). We believe the increased
underestimation during polluted conditions is likely due to enhanced condensation of secondary hygroscopic
compounds (e.g. nitrate, sulfate) on pre-existing aerosols at lower temperature and higher relative humidity
at nighttime (Wu et al., 2008; Wang et al., 2016; An et al., 2019).

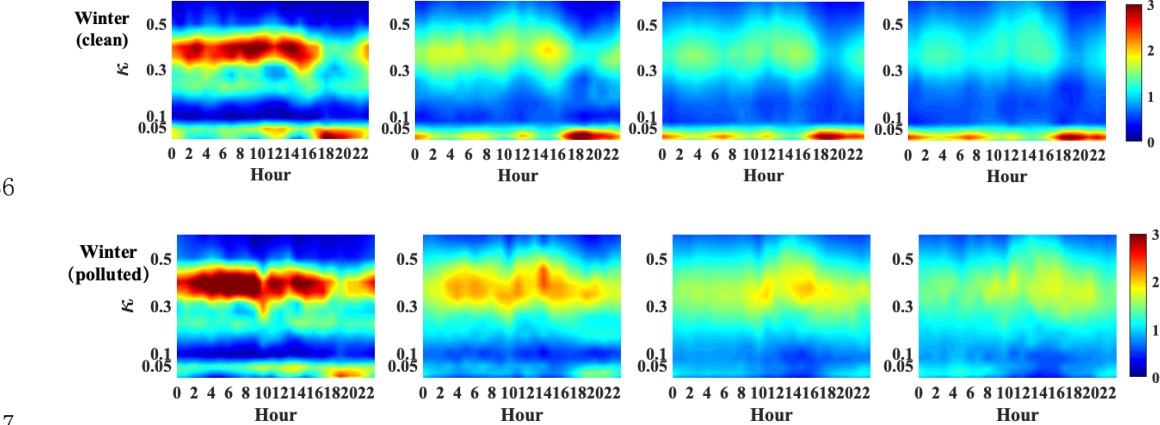



Figure 8. Diurnal cycles of $\kappa_{gf}$ -PDF for 80, 110, 150 and 200 nm particles in clean and polluted events in
winter.

**3.5. Observation from other stations**

The aging process in the summer period is related to photochemical processing in strong solar radiation
conditions. The photochemical reactions produce sulfate and secondary organic aerosol, condensing on the
surface of slightly- or non-hygroscopic primary aerosols (such as black carbon) (Zhang et al., 2008). As
discussed in 3.4, the core-shell structure that accompanies aging of the particles results in calculated $\kappa_{chem}$
that underestimates their hygroscopicity. To confirm such a coating effect on particle hygroscopicity, we
further examine the diurnal variations of $\kappa_{chem}$ and $\kappa_{gf}$ or $\kappa_{CCNc}$ (derived from CCN measurements) based on
observations in summer at two other sites in north China (Fig. 1). We find that the case at the Xingtai (XT)
site is very similar to that observed in Beijing (BJ), with a lower $\kappa_{chem}$ than $\kappa_{gf}$ around noon time. But,
because of much less influences from the local sources at XT compared to that at BJ, such underestimation
by $\kappa_{chem}$ continued until night at XT (Fig. 9b). Interestingly, a noontime lower $\kappa_{chem}$ was not observed in the
diurnal cycles at the Xinzhou (XZ) site, where $\kappa_{chem}$ and $\kappa_{CCNc}$ had similar diurnal patterns (Fig. 9c) with a
roughly constant ratio of $\kappa_{chem}$ to $\kappa_{CCNc}$ of ~0.8-0.9. This is probably because the XZ site is usually the
recipient of aerosols transported from other areas that are already aged and well-mixed, with minimal impact
of additional coating (Zhang et al., 2017). Also, the rate of oxidation and condensation may be slow in the
relatively remote area where the gas precursors and oxidants are not as high as they are closer to sources
regions. But at XT, which is located in the heavily polluted area in the north China Plain (Fu et al., 2014),
aerosol emissions and processing are more similar to that in urban Beijing.

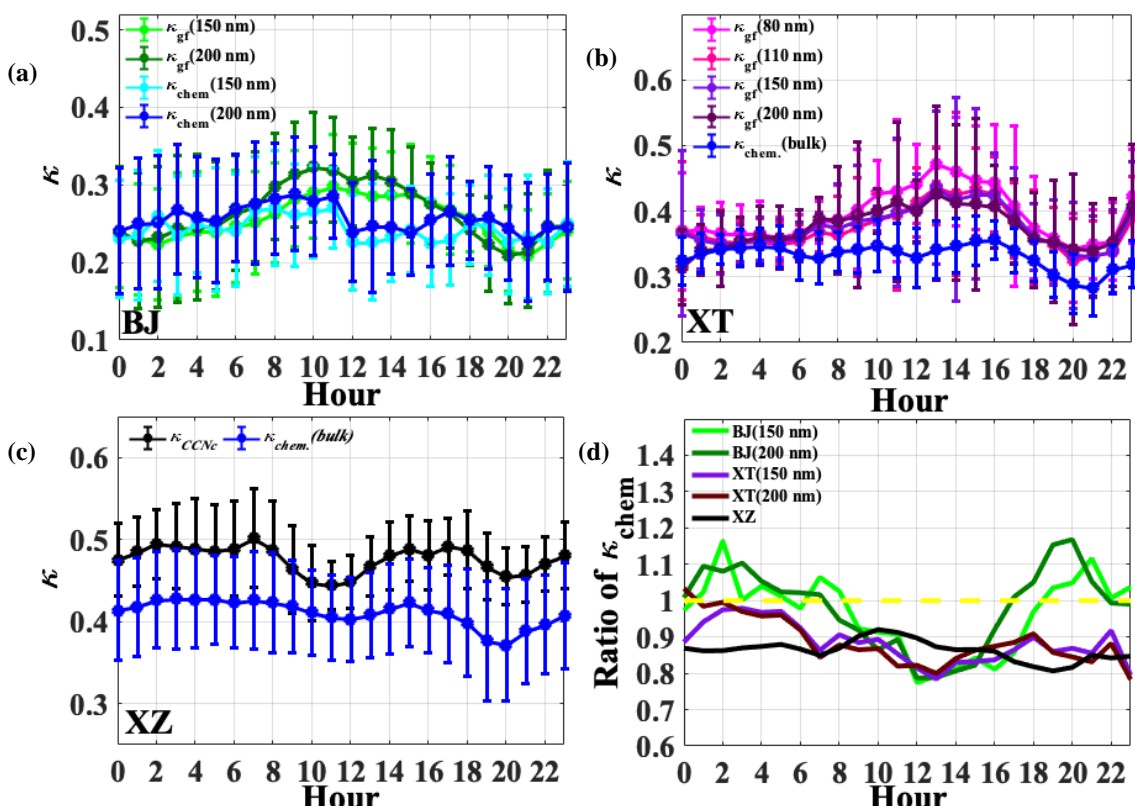


Figure 9. Diurnal variations in (a) $\kappa_{chem}$ and $\kappa_{gf}$ for 150 and 200 nm particles at BJ site; (b) $\kappa_{chem}$ and $\kappa_{gf}$ for
40, 80, 110, 150 and 200 nm particles at XT site; (c) $\kappa_{chem}$ and mean $\kappa_{CCNc}$ for particles at XZ site, and (d)
ratio of mean $\kappa_{chem}$ to $\kappa_{gf}$ at the three sites.

Although the underestimation in $\kappa_{chem}$ may be also related to the uncertainty in the hygroscopic

parameter for organics, which is calculated from a simple parametrized equation based on the
AMS-measured $f_{44}$ value reported by Mei et al. (2013), Zhang et al. (2017) has shown that even the large





365 underestimation of $\kappa_{SOA}$ could not fully explain that of $\kappa_{chem}$. Furthermore, the value for $f_{44}$ tends to be

366 overestimated according to Fröhlich et al. (2015), which should lead to a larger $\kappa_{chem}$. Previous studies have

367 shown that freshly emitted POA and BC particles may be rapidly coated by more hygroscopic components

368 in polluted urban areas, resulting in enhanced hygroscopicity of the mixed particles (Zhang et al., 2004;

369 Johnson et al., 2005; Zhao et al., 2017). Our results are consistent with those observations and clarify the

370 photochemical aging and coating effect will largely underestimate the particles hygroscopisity using simple

371 mixing rule based on chemical composition.

**4. Conclusion**

373 Using measurements of aerosol composition and hygroscopicity made in Beijing (BJ) during a winter

374 period of 2016 and a summer period of 2017, this paper analyzes the daily variation and seasonal differences

375 of size-resolved aerosol hygroscopicity in urban Beijing. We mainly focus on studying the disparity of $\kappa_{gf}$

376 and $\kappa_{chem}$ between summer and winter to reveal the impact of atmospheric processes and mixing state of the

377 particles on its hygroscopicity. The uncertainty in calculating $\kappa$ by using chemical composition with a

378 uniform internal mixing hypothesis is elucidated from the diurnal variations of the difference between the

379 calculated values: in summer, lower $\kappa_{chem}$ is obtained around noontime, with a ratio of $\kappa_{chem}$ to $\kappa_{gf}$ of about

380 0.8-0.9 for large particles (i.e. 150 nm and 200 nm), showing an underestimation of particles hygroscopisity

381 by using simple mixing rule based on chemical composition. Combining with the observation from Xingtai

382 and Xinzhou, we attribute the underestimation to the rapid noontime photochemical aging processes in

383 summer, which induces the coating effect that will lead to a lower $\kappa$ if assuming a uniform mixing of the

384 particles. In contrast, larger $\kappa_{chem}$ than $\kappa_{gf}$ for >100 nm particles around noontime and in the early afternoon

385 is derived in winter, with the maximum ratio of $\kappa_{chem}$ to $\kappa_{gf}$ of 1.2-1.4 when the particles are dominated by

386 the hydrophobic mode with a large number of externally-mixed POA particles from strong vehicle and

387 cooking sources. We suggest that, by using the simple mixing rule, the particles hygroscopisity can be

388 underestimated up to 10%-20% for aged aerosols due to the coating effect, but will be maximally

389 overestimated 20-40% for externally-mixed particles. A lower $\kappa_{chem}$ than $\kappa_{gf}$ for 80, 110 and 150 nm particles

390 during the nighttime of winter is also noted, and the disparity is further enlarged in polluted days, probably





due to a nighttime coating effect driven by condensation of secondary hygroscopic species on pre-existing
aerosols in cold season. Our results highlight the impacts of atmospheric processes, sources on aerosol
mixing state and hygroscopicity, which should be quantified and considered in models for different
atmospheric conditions. Long-term observations from more ground sites, as well as experiments in smog
chambers, should be made to parameterize such impact in model simulations.

*Data availability.* All data needed to evaluate the conclusions in the paper are present in the paper and/or the
Supplementary Materials. Also, all data used in the study are available from the corresponding author upon
request (fang.zhang@bnu.edu.cn).
*Author contributions.* F.Z. and X.F. conceived the conceptual development of the manuscript. X. F. directed
and performed of the experiments with L.C., X.J., Y. W., and F. Z.. X.F. and F.Z. conducted the data
analysis and wrote the draft of the manuscript, and all authors edited and commented on the various sections
of the manuscript.
*Competing interests.* The authors declare no competing interests.
*Acknowledgements.* This work was funded by the National Key R&D Program of China (grant no.
2017YFC1501702), National Natural Science Foundation of China (NSFC) research projects (grant nos.
41675141, 91544217). We thank all participants of the field campaign for their tireless work and
cooperation.

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
