# Peer review of "Contrasting size-resolved hygroscopicity of fine particles derived by HTDMA and HR-AMS 1"

_Atmospheric Chemistry and Physics, 2019_

## Referee Comment (RC1) · Anonymous Referee #1 · 11 Sep 2019

In this manuscript in discussion for publication in Atmospheric Chemistry and Physics (acp-2019-583), Xinxin Fan and co-authors present a field study comparing aerosol hygroscopicity in summer months relative to the those measured in winter. Measured hygroscopicity was compared to hygroscopicity based on HR-ToF-AMS measurements of composition for Beijing and northern China. The focus on this work was mixing state as a potential cause of the discrepancy between measured and estimated hygroscopicity. Interesting observations are presented and discussed in a mechanistic framework. This work is part of a larger effort to understand the air quality in China, and is important and timely. I have significant concerns, however, about the novelty of the study and the presentation of the data, which I have outlined below. The data and study design are not novel, and in fact several of the same authors have written a very similar manuscript (published in ACP: https://www.atmos-chem-phys.net/18/11739/2018/acp-18-11739-2018.pdf) from the same field campaign. The preparation of figures as clear and succinct visual aids to the writing is poor, and the authors invoke limited and dated studies on water uptake by mixtures of compounds. These issues could potentially be resolved with appropriate major revisions.

Regarding the novelty of the manuscript, I would urge the authors to share in the introduction the previous findings for the same dataset or the co-located instruments. It is not clear at present the degree of overlap but it is not the policy of ACP to publish the same data, analysis, and interpretation twice. The difference between (for example) the CCN and HTDMA needs to be clearly stated in both the method and the interpretation and discussion of underlying physical processes. If the authors do not differentiate effectively between the scientific questions answered by similar instruments, then the study is essentially the same as the published study. This can likely be resolved but will require careful effort.

Comments on figures and interpretation of figures:

The figures do not always serve as appropriate and helpful guides to the writing. The number of figures in both the manuscript and the supplement could be reduced. Not all figures are discussed, and several figures seem to be entirely redundant. The data in the figures is difficult to interpret due to the overlapping error bars.

Figure 3: It's not clear why this figure does not take the full page width, as it already seems to exceed a 1-column width. It would be helpful to include markers for "morning traffic," "afternoon traffic," or other factors that influence these timeseries. The reader is without a frame of reference. Also, in the caption it would be helpful to see the location for these timeseries, or whether these are averaged for all sites.

Line 218: Figure 3e is referenced before any discussion of all the other panels in Figure 3.

Figure 5: Authors neglect to describe the two lines on each plot; are the R2 values first or second in the parentheses? Are the 1:1 lines anchored at 0? There seems to be little to no correlation between kchem and kgf.

Line 275: These numbers don't match the figure. With R2 values of 0.01-0.23 for the kchem vs kgf correlations, I would hesitate to report the slope of the fit line. Anchoring the line and a value other than (0,0) would give a different slope with a similar R2 value.

Line 292: In figure 6 the gap between kgf and kchem for larger particles looks similar across all plots. A closer look that kchem is higher in the late afternoon only in winter, and lower in summer. But, all the error bars appear to overlap almost completely. I strongly recommend displaying the data such that the error bars can be distinguished. By way of example: the dotted lines in the background are unhelpful, the resolution of the figure is not high, and the midpoint of the error bar is not entirely necessary if the error bars are symmetric above/below this point. Some authors use overlapping shaded regions (https://andrewpwheeler.wordpress.com/2016/03/08/on-overlapping-error-bars-in-charts/).

In panel B the yellow trace is hard to see. Error bars are omitted.

Figure S6: How is Figure S6 different from Figure 6?

Figure S1 and others: Kappa should not be negative and this could indicate evaporation of some fraction of particles.

Comments on underlying physical processes

The readership may already have an understanding of internal vs external mixtures. The description of internal vs external mixing is not succinct and does not contain many references – I suggest reducing the length of this review and incorporating the following elements: more quantitative information, more references and conclusions drawn from previous work.

Line 53: Are they?

Water uptake by coated particles (including those coated with aliphatic compounds) is likely not inhibited (https://www.pnas.org/content/110/22/8807, https://www.atmos-chem-phys.net/19/3325/2019/acp-19-3325-2019.html).

Line 71-73: There have been continuing studies of the hygroscopicity of mixed aerosols under controlled conditions, which may provide additional framework for mechanistic discussion:

https://pubs.acs.org/doi/full/10.1021/acscentsci.5b00174

https://agupubs.onlinelibrary.wiley.com/doi/full/10.1029/2011JD016823

https://agupubs.onlinelibrary.wiley.com/doi/full/10.1029/2007JD009274

https://pubs.acs.org/doi/10.1021/acs.jpca.5b09373

---

## Referee Comment (RC2) · Anonymous Referee #2 · 27 Oct 2019

In this manuscript, Fan et al. measured the hygroscopicity and chemical composition of the size-resolved aerosols at several locations in northern China, and calculated the hygroscopic parameter ($\kappa$) based on both the hygroscopic growth factor from HT-DMA measurement ($\kappa\_gf$) and the chemical composition from HR-AMS measurement ($\kappa\_chem$). By comparing $\kappa\_gf$ and $\kappa\_chem$, this study demonstrates clear and undisputed evidence of possible bias in estimating aerosol hygroscopicity using the chemical mixing rule. Moreover, Fan et al. provides reasonable insight on the influence of atmosphere process and aerosol mixing state on the calculation of aerosol hygroscopicity. The manuscript is well organized and written. I will recommend the publication of this manuscript in ACP, as long as the following comments are properly addressed. Note

that comments 4-6 are just suggestions. (1) A major discovery of the paper is that $\kappa$_chem calculated using the mixing rule cannot reflect the aerosol hygroscopicity. For example, it is found that the $\kappa$_chem in summer is underestimated at noon, overestimated at late peak hours, and substantially consistent with k_gf at midnight. Though I think the results should be correct, I am not fully convinced by some of the interpretation. (a) Why the external mixing of BC and POA with other components during the late peak hour will result in overestimation of $\kappa$_chem? (b) According to the author's argument, aerosols both at noon and at midnight have core-shell structure, but why the $\kappa$_chem/k_gf is quite distinct? More detailed interpretation and discussion are necessary. (2) L259, "Since a size-resolved BC mass concentration measurement was not available during the campaign, we use the bulk mass fraction of BC particles measured by the AE33 combining with size-resolved BC distribution in Beijing reported by Liu et al. (2018) to estimate $\kappa$chem." As far as I know, the instrument to measure the size distribution of BC in Liu et al. (2018) is a SP2, which gives the BC core diameter. It is necessary to explain how to convert this size distribution of BC core to the size distribution of ambient aerosols. (3) L227 and fig. 3. "the concentration of the hydrophilic mode increased quickly around noontime and in the early afternoon (12:00-16:00)", which is explained by a transformation of the particles from externally to internally mixing state. However, I have different opinion. From Fig. 3a, it is evident that 40 nm particles after 12:00 were dominated by new particle formation (NPF). Therefore, the decrease of hydrophobic mode could be attribute to the extremely large amount of hydrophilic particles from NPF overwhelmed all other particles. (4) It will be better if the authors can discuss more on the similarities and differences of the hygroscopicity calculation at different sites. (5) There have been several studies revealing the uncertainty of calculating hygroscopicity using the mixing rule, but few can provide proper solution. Is it possible for the authors to propose parameterized modification on the $\kappa$_chem to reduce the uncertainty? If so, this paper will be enormously improved and will be far distinct from other studies. For example, should we use lower BC density value during the rush hours? (6) For several times, the current manuscript cited Zhang

et al. (2017), which is one of the previous studies done by the same group on the same topic. Therefore, it is appropriate to make a clear statement of the unresolved issues in the previous paper or what improvement has been made to this study so that the reader can easily understand the novelty of this paper.

Other minor comments: (1) fig. 2 is not reader-friendly. Please work out some way to make the information more clear. (2) fig.3. There are totally 12 sub-figures here. Please consider naming each sub-figures rather than the current way (which is not clearly demonstrated). (3) L150 and L160, the full term and the abbreviations of probability density functions (PDF )should be provided the first time in the text. (4) Fig. 5, L266, should be "slopes of linear fits and correlation coefficients".
* * *

---

## Author Comment (AC1) · 22 Nov 2019

Anonymous Referee #1 In this manuscript in discussion for publication in Atmospheric Chemistry and Physics (acp-2019-583), Xinxin Fan and co-authors present a field study comparing aerosol hygroscopicity in summer months relative to the those measured in winter. Measured hygroscopicity was compared to hygroscopicity based on HR-ToF-AMS measurements of composition for Beijing and northern China. The focus on this work was mixing state as a potential cause of the discrepancy between measured and estimated hygroscopicity. Interesting observations are presented and discussed in a mechanistic framework. This work is part of a larger effort to understand

the air quality in China, and is important and timely. I have significant concerns, how-ever, about the novelty of the study and the presentation of the data, which I have out-lined below. The data and study de-sign are not novel, and in fact several of the same authors have written a very similar manuscript (published in ACP: https://www.atmos-chem-phys.net/18/11739/2018/acp-18-11739-2018.pdf) from the same field campaign. The preparation of figures as clear and succinct visual aids to the writing is poor, and the authors invoke limited and dated studies on water uptake by mixtures of com-pounds. These issues could potentially be resolved with appropriate major revisions. Regarding the novelty of the manuscript, I would urge the authors to share in the intro-duction the previous findings for the same dataset or the co-located instruments. It is not clear at present the degree of overlap but it is not the policy of ACP to publish the same data, analysis, and interpretation twice. The difference between (for example) the CCN and HTDMA needs to be clearly stated in both the method and the interpreta-tion and discussion of underlying physical processes. If the authors do not differentiate effectively between the scientific questions answered by similar instruments, then the study is essentially the same as the published study. This can likely be resolved but will require careful effort.

Re: We appreciate your comments. The reviewer argued that the paper published in ACP and this currently submitted one is very similar manuscript from the same field campaign. This is probably because that some vague descriptions on instruments in the Section 2.1 which may have mislead the reviewer. Indeed, the main data used in the two papers are from different campaigns, the data used in this work are from two field campaigns during November 16-December 10 of 2016 and May 25- June 18 of 2017 in urban Beijing, however, the published ACP paper just used the data from Xingtai campaign which was conducted during 1 May-15 June 2016. These have been clarified in the revised manuscript (See lines 92-98, 383-393). Furthermore, the pre-vious paper published in ACP focused on investigating and characterizing the aerosol hygroscopicity and CCN activity at the suburban site of Xingtai, which is located about 420 km south of urban Beijing. But in the current submitted paper, we compare the

size-resolved hygroscopic parameter ($\kappa$gf) of ambient fine particles derived by an HT-DMA (Hygroscopic Tandem Differential Mobility Analyzer) to that (denoted as $\kappa$chem) of calculated by an HR-ToF-AMS (High-resolution Time-of-Flight Aerosol Mass Spectrometer) measurements using a simple rule with a uniform internal mixing hypothesis. We mainly focus on contrasting the disparity of $\kappa$gf and $\kappa$chem between summer and winter in urban Beijing to reveal the impact of atmospheric processes/sources on aerosols hygroscopicity and to evaluate the uncertainty in estimating particles hygroscopicity with the hypothesis. Only in the last section (Section 3.5) of this paper, we include the observations at other sites (not only Xingtai site) just for comparison with that observed in urban Beijing. Such comparison among different sites is to identify the impact of regional emissions/sources and atmospheric processes under different environments on estimating aerosols hygroscopisity with the uniform internal mixing hypothesis. One important findings of this current paper is that, for the first time, we observe clearly that atmospheric photochemical aging of aerosols induces a coating effect from field measurement. Such effect leads to 10%-20% underestimation of the hygroscopic parameter if using the uniform internal mixing assumption. The coating effect is found more significant for these >100 nm particles observed in remote or clean regions. Our results suggest that it is critical to parameterize such an impact in model simulations to improve the evaluation of the aerosols indirect effect. In addition, in the revised version, we have made a sensitivity test to examine the effect of temporal variations in actual density of BC and organics caused by the particles aging and local sources on calculating $\kappa$chem (see lines 319-346). The figures have been revised carefully according to the comments (see the revised Fig. 1-Fig. 9). In addition, more previous studies and references on water uptake by mixtures of compounds have been included in the introduction, and some words about the definition of mixing state have been removed in the revised version. The revised introduction is as follows, " . . .The hygroscopic properties of both the natural and anthropogenic aerosols, in addition to being affected by its chemical composition (Gunthe et al., 2009), are also affected by the particle mixing state and aging (Schill et al., 2015; Peng et al., 2017). For example, a recent laboratory study shown that the coexisting hygroscopic species have a strong influence on the phase state of particles, thus affecting chemical interactions between inorganic and organic compounds as well as the overall hygroscopicity of mixed particles (Peng et al., 2016). The field measurements also demonstrated that the hydrophobic black carbon particles became hygroscopic with atmospheric mixing and aging by organics (i.e. Peng et al., 2017). In a heavily polluted atmosphere, the aerosol sources and sinks are varied, the physical and chemical processes experienced by the aerosols are complex, and the mixing state and its impact on aerosols hygroscopicity is more complicated. The hygroscopicity of mixed particles and mutual impacts between the components are still poorly understood. Previous studies have shown that the difference between the $\kappa$ obtained from H-TDMA or CCNc measurements and that calculated based on the volume mixing ratio of chemical components, $\kappa$chem. Laboratory results from Cruz and Pandis (2000) indicate that $\kappa$gf of internally mixed ammonium sulfate and organic matter is higher than $\kappa$chem calculated for assumed uniform internal mixing. But Peng et al (2016) found that, for sodium chloride and organic aerosols mixed particles, the measured growth factors by H-TDMA were significantly lower than calculations from the mixing rule methods. In some field studies on aged aerosols, the $\kappa$ was underestimated by the calculation based on uniform internal mixing assumption and thus lead to an underestimation of CCN concentration(Bougiatioti, et al., 2009; Chang, et al., 2007; Kuwata, et al., 2008; Wang, et al., 2010; Ren et al., 2018). However, for primary emissions dominated periods, the $\kappa$ value from calculations based on bulk chemical composition was much higher than that measured by H-TDMA measurements (Zhang et al., 2017). The various results from previous studies suggest distinct effects of aerosols mixing state on their hygroscopicity. Overall, to what extent do the differences depend on the mixing state and the extent of aging of the particles, and how the different atmospheric processes and what kinds of mixing structure of the particles may result in those disparity between measured and calculated hygroscopic parameter have not been clearly clarified by the previous studies. A comprehensive and systematic investigation on the cause and

magnitude of the effect has been lacking. In the atmosphere, the $\kappa$, which is related to the particle mixing state diversity, varies largely across the size range of ambient fine particles (Rose et al., 2010). Previous study only compared the measured$\kappa$to that calculated based on bulk chemical composition (Zhang et al., 2017). Using size-resolved, not bulk, chemical composition measurements in different seasons is expected to provide more comprehensive understanding and insights of how the aerosols mixing state influence on their hygroscopicity, motivating our analysis that employs size-resolved chemical composition measured by an HR-ToF-AMS in this study.…." The difference between (for example) the CCNc and HTDMA has been stated in the revised version (see lines 179-189) or as follows, "…In addition, we also compare the results from the field campaigns with those from other two sites, Xingtai (XT: 37.18° NïijŇ114.37° E), and Xinzhou (XZ: 38.24° NïijŇ112.43° E), in North China Plain (Fig. 1). At XZ site, we use the hygroscopic parameter (defined as $\kappa$CCNc) from size-resolved CCN measurements (Zhang et al., 2014, 2016) for comparison. More detailed descriptions of the method to retrieve $\kappa$CCNc can be found in (Petters and Kreidenweis (2007). Both of the $\kappa$gf and $\kappa$CCNc are derived based on $\kappa$ -KoÌĹhler Theory (Petters and Kreidenweis, 2007). But, different from the $\kappa$gf measured by the HTDMA system which is operated at RH of 90%, the $\kappa$CCNc is derived by measuring aerosols CCN activity under the condition of supersaturations with relative humidity of >100%. Previous studies from filed measurements and laboratory experiments showed that the $\kappa$CCNc is generally slight larger or smaller than $\kappa$gf, but they are basically comparable and can well represent an overall aerosols hygroscopisity (e.g. Carrico et al., 2008; Wex et al., 2009; Good et al., 2010; Irwin et al., 2010; Cerully et al., 2011; Wu et al., 2013; Zhang et al., 2017).…."

Comments on figures and interpretation of figures: The figures do not always serve as appropriate and helpful guides to the writing. The number of figures in both the manuscript and the supplement could be reduced. Not all figures are discussed, and several figures seem to be entirely redundant. The data in the figures is difficult to interpret due to the overlapping error bars.

Re: As commented by the reviewer, we have considered how to organize the figures very carefully, and removed most of the figures in both the main manuscript and the supplement in the revised version. In addition, the Figures in the main text were replotted due to the overlapping error bars (see the revised Fig. 1-Fig. 9).

Figure 3: It's not clear why this figure does not take the full page width, as it already seems to exceed a 1-column width. It would be helpful to include markers for "morning traffic," "afternoon traffic," or other factors that influence these timeseries. The reader is without a frame of reference. Also, in the caption it would be helpful to see the location for these time series, or whether these are averaged for all sites.

Re: The figure and caption have been revised per the reviewer's comments (see below),

Figure R 1. Campaign averaged diurnal variations in particle number size distribution; mass concentration of PM1, bulk mass concentration of main species in PM1, mass fraction of chemical composition of PM1; and Gf-PDFs for 40 and 150 nm particles in winter (left panels) and summer (right panels) measured in urban Beijing. Line 218: Figure 3e is referenced before any discussion of all the other panels in Figure 3.

Re: The Fig.3 has been mentioned in the previous paragraph before line 218. However, corresponding revision of the text has been done according to the correction on Fig. 3.

Figure 5: Authors neglect to describe the two lines on each plot; are the R2 values first or second in the parentheses? Are the 1:1 lines anchored at 0? There seems to be little to no correlation between $\kappa$chem and $\kappa$gf.

Re: Thanks a lot for the careful check. In the revised version, we have added the description about two lines. The first number in parenthesis of each plot is the slope of the fit line, and the second is the correlation coefficient (R2). In figure 5, all 1:1 lines are anchored at 0. Exactly, the correlations between $\kappa$chem and $\kappa$gf of the 80, 110, 150, 200 nm particles both in winter and summer are poor due to the large uncertainty

in one or both of the calculated parameters. The large uncertainties are likely due to the unreasonable assumption of particle mixing state, which varies with their aging and other physiochemical processes in the atmosphere. This has been stated in the text.

Line 275: These numbers don't match the figure. With R2 values of 0.01-0.23 for the $\kappa$chem and $\kappa$gf correlations, I would hesitate to report the slope of the fit line. Anchoring the line and a value other than (0,0) would give a different slope with a similar R2 value.

Re: Yes, the reviewer is right. The discussion about the slopes and R2 has been revised (See lines 272-280) as follows, ""...The results show that, although the slopes from linear fitting of $\kappa$chem and $\kappa$gf are close to 1.0, it is with quite poor ccorrelations (typically with correlation coefficients, R2, of < 0.3) between $\kappa$chem and $\kappa$gf of the 80, 110, 150, 200 nm particles both in winter and summer. The poor correlations reflect large uncertainty in one or both of the calculated parameters that are likely due to the unreasonable assumption of particle mixing state (e.g. Cruz and Pandis, 2000; Svenningsson et al., 2006; Sjogren et al., 2007; Zardini et al., 2008), which varies with their aging and other physiochemical processes in the atmosphere. Note that underestimation of $\kappa$chem for the summer occurred mostly in the afternoon (Marked in blue dots in Fig. 5). This may be associated with photochemical processes at around noontime. More specific investigations of the particle mixing and aging impacts on $\kappa$chem will be further addressed in the following sections...."

Line 292: In figure 6 the gap between $\kappa$gf and $\kappa$chem for larger particles looks similar across all plots. A closer look that $\kappa$chem is higher in the late afternoon only in winter, and lower in summer. But, all the error bars appear to overlap almost completely. I strongly recommend displaying the data such that the error bars can be distinguished. By way of example: the dotted lines in the background are unhelpful, the resolution of the figure is not high, and the midpoint of the error bar is not entirely necessary if the error bars are symmetric above/below this point. Some authors use overlapping shaded regions. In panel B the yellow trace is hard to see. Error bars are omitted.

Re: Thanks for the comments. The figure has been revised. As the reviewer suggested, we use shaded regions to indicate the error bar (see Fig. R2).

Figure S6: How is Figure S6 different from Figure 6?

Re: To examine the impacts of pollution conditions on the diurnal variations of $\kappa$, Figure S6 (Now Fig S1 in the revised version) shows the diurnal cycles under clean and polluted cases respectively in winter; while Fig 6 just shows an overall diurnal change of $\kappa$ in summer and winter.

Figure S1 and others: Kappa should not be negative and this could indicate evaporation of some fraction of particles.

Re: These figures have been revised (see an example as follows, Fig. R3). But is was removed from the revised version according to reviewer's comments.

Comments on underlying physical processes The readership may already have an understanding of internal vs external mixtures. The description of internal vs external mixing is not succinct and does not contain many references – I suggest reducing the length of this review and incorporating the following elements: more quantitative information, more references and conclusions drawn from previous work.

Re: More previous studies and references on water uptake by mixtures of compounds have been included in the introduction, and some words about the definition of mixing state have been removed in the revised version (Lines 50-82) as follows, "…The hygroscopic properties of both the natural and anthropogenic aerosols, in addition to being affected by its chemical composition (Gunthe et al., 2009), are also affected by the particle mixing state and aging (Schill et al., 2015; Peng et al., 2017). For example, a recent laboratory study shown that the coexisting hygroscopic species have a strong influence on the phase state of particles, thus affecting chemical interactions between inorganic and organic compounds as well as the overall hygroscopicity of mixed particles (Peng et al., 2016). The field measurements also demonstrated that

the hydrophobic black carbon particles became hygroscopic with atmospheric mixing and aging by organics (i.e. Peng et al., 2017). In a heavily polluted atmosphere, the aerosol sources and sinks are varied, the physical and chemical processes experienced by the aerosols are complex, and the mixing state and its impact on aerosols hygroscopicity is more complicated. The hygroscopicity of mixed particles and mutual impacts between the components are still poorly understood. Previous studies have shown that the difference between the $\kappa$ obtained from H-TDMA or CCNc measurements and that calculated based on the volume mixing ratio of chemical components, $\kappa$chem. Laboratory results from Cruz and Pandis (2000) indicate that $\kappa$gf of internally mixed ammonium sulfate and organic matter is higher than $\kappa$chem calculated for assumed uniform internal mixing. But Peng et al (2016) found that, for sodium chloride and organic aerosols mixed particles, the measured growth factors by H-TDMA were significantly lower than calculations from the mixing rule methods. In some field studies on aged aerosols, the $\kappa$ was underestimated by the calculation based on uniform internal mixing assumption and thus lead to an underestimation of CCN concentration(Bougiatioti, et al., 2009; Chang, et al., 2007; Kuwata, et al., 2008; Wang, et al., 2010; Ren et al., 2018). However, for primary emissions dominated periods, the $\kappa$ value from calculations based on bulk chemical composition was much higher than that measured by H-TDMA measurements (Zhang et al., 2017). The various results from previous studies suggest distinct effects of aerosols mixing state on their hygroscopicity. Overall, to what extent do the differences depend on the mixing state and the extent of aging of the particles, and how the different atmospheric processes and what kinds of mixing structure of the particles may result in those disparity between measured and calculated hygroscopic parameter have not been clearly clarified by the previous studies. A comprehensive and systematic investigation on the cause and magnitude of the effect has been lacking. In the atmosphere, the $\kappa$, which is related to the particle mixing state diversity, varies largely across the size range of ambient fine particles (Rose et al., 2010). Previous study only compared the measured$\kappa$to that calculated based on bulk chemical composition (Zhang et al., 2017). Using size-resolved,

not bulk, chemical composition measurements in different seasons is expected to pro-
vide more comprehensive understanding and insights of how the aerosols mixing state
influence on their hygroscopicity, motivating our analysis that employs size-resolved
chemical composition measured by an HR-ToF-AMS in this study.. . .."

Line 53: Are they? Water uptake by coated particles (including those coated with
aliphatic compounds) is likely not inhibited. Re: This should be ". . . . . .In the case of
external mixing, the chemical components in the aerosol particles are independent of
each other, and the chemical composition of the different types of aerosol particles
is different within a certain particle size range." However, we have made a through
revision of the introduction part.

Line 71-73: There have been continuing studies of the hygroscopicity of mixed
aerosols under controlled conditions, which may provide additional framework for
mechanistic discussion. https://pubs.acs.org/doi/full/10.1021/acscentsci.5b00174
https://agupubs.onlinelibrary.wiley.com/doi/full/10.1029/2011JD016823
https://agupubs.onlinelibrary.wiley.com/doi/full/10.1029/2007JD009274
https://pubs.acs.org/doi/10.1021/acs.jpca.5b09373

Re: We really appreciate your comments. These studies above listed are very helpful
for improving our understanding of hygroscopicity of mixed aerosols. More discussions
about the effect of mixed aerosols on hygroscopisity have been included in the revised
manuscript by referring these studies in both the introduction, method and the inter-
pretation and discussion of underlying physical processes. For example, Lines 50-55,
"The hygroscopic properties of both the natural and anthropogenic aerosols, in addition
to being affected by its chemical composition (Gunthe et al., 2009), are also affected by
the particle mixing state (Schill et al., 2015; Peng et al., 2017). For example, a recent
laboratory study shown that the coexisting hygroscopic species have a strong influence
on the phase state of particles, thus affecting chemical interactions between inorganic
and organic compounds as well as the overall hygroscopicity of mixed particles (Peng
et al., 2016). The field measurements also demonstrated that the . . ..."

Lines 61-66, "Previous studies have shown that the difference between the $\kappa$ obtained from H-TDMA or CCNc measurementsand that calculated based on the volume mixing ratio of chemical components, $\kappa$chem. Laboratory results from Cruz and Pandis (2000) indicate that $\kappa$gf of internally mixed ammonium sulfate and organic matter is higher than $\kappa$chem calculated for assumed uniform internal mixing. But Peng et al (2016) found that, for sodium chloride and organic aerosols mixed particles, the measured growth factors by H-TDMA were significantly lower than calculations from the mixing rule methods. In some field studies on aged aerosols,"

Lines 184-189, "....But, different from the $\kappa$gf measured by the HTDMA system which is operated at RH of 90%, the $\kappa$CCNc is derived by measuring aerosols CCN activity under the condition of supersaturations with relative humidity of >100%. Previous studies from filed measurements and laboratory experiments showed that the $\kappa$CCNc is generally slight larger or smaller than $\kappa$gf, but they are basically comparable and can well represent an overall aerosols hygroscopisity (e.g. Carrico et al., 2008; Wex et al., 2009; Good et al., 2010; Irwin et al., 2010; Cerully et al., 2011; Wu et al., 2013; Zhang et al., 2017)....."

Lines 351-355, "...Besides the impacts of BC aging (changes in morphology/density) and variations of the overall density of organics on particles hygroscopicity, uncertainty in $\kappa$chem may be related to the uncertainty in the hygroscopic parameter for organics that could vary widely over a range of diverse constitutes of SOA (Suda et al., 2012). However, Zhang et al. (2017) shown that using a smaller or larger $\kappa$SOA could not fully explain the overestimation during traffic hours or the underestimation around noontime...."

Please also note the supplement to this comment:
https://www.atmos-chem-phys-discuss.net/acp-2019-583/acp-2019-583-AC1-supplement.pdf

---

## Author Comment (AC2) · 22 Nov 2019

Anonymous Referee #2 In this manuscript, Fan et al. measured the hygroscopicity and chemical composition of the size-resolved aerosols at several locations in northern China, and calculated the hygroscopic parameter ($\kappa$) based on both the hygroscopic growth factor from HTDMA measurement ($\kappa$_gf) and the chemical composition from HR-AMS measurement ($\kappa$_chem). By comparing $\kappa$_gf and $\kappa$_chem, this study demonstrates clear and undisputed evidence of possible bias in estimating aerosol hygroscopicity using the chemical mixing rule. Moreover, Fan et al. provides reasonable insight on the influence of atmosphere process and aerosol mixing state on the calculation

of aerosol hygroscopicity. The manuscript is well organized and written. I will recommend the publication of this manuscript in ACP, as long as the following comments are properly addressed. Note that comments 4-6 are just suggestions.

Re: We are grateful to reviewer 2 for the insightful and constructive comments and have revised our paper accordingly to account for the reviewer's recommendations.

(1) A major discovery of the paper is that $\kappa$chem calculated using the mixing rule cannot reflect the aerosol hygroscopicity. For example, it is found that the $\kappa$chem in summer is underestimated at noon, overestimated at late peak hours, and substantially consistent with kgf at midnight. Though I think the results should be correct, I am not fully convinced by some of the interpretation. (a) Why the external mixing of BC and POA with other components during the late peak hour will result in overestimation of $\kappa$chem?

Re: The emission of a large number of primary hydrophobic particles like BC and POA leads to great decrease of the overall aerosol hygroscopicity. We have included discussions and statements in the revised manuscript (lines 319-346, Fig. 8) as follows (Fig. R1), "...We suppose that the large disparity between $\kappa$chem and $\kappa$gf is due to temporal variations in actual density of BC and organics caused by the particles aging and local sources. The externally-mixed BC particles are with fractal structure and chain-like aggregates and have been reported with effective density of 0.25-0.45 g cm-3(McMurry et al., 2002), While the BC particles in the calculation is assumed as void free with effective density of 1.7 g cm-3. Such inappropriate assumption would lead to an underestimation of BC volume fraction and thus the overestimation in $\kappa$chem during the traffic rush hour and cooking time when BC particles are mostly freshly emitted with uncompacted structure. In addition, the significant increase in volume fraction of POA during the late afternoon would result in a lower density of organics, which is expected to be smaller than the assumed one (1.2 g cm-3) in the calculation. A sensitivity test has been done to examine the effect of density of BC and organics on calculated $\kappa$chem (Fig. 7). The result shows that the $\kappa$chem value reduces by 16-33% when applying the BC effective density of 0.25-0.45 g cm-3. This basically explains the

disparity during the traffic rush hour. However, the changes in $\kappa$chem are within $\pm4\%$ when changing the organic density from 1.0 (typical for POA) to 1.4 (typical for SOA) g cm-3, suggesting insensitivity of $\kappa$chem to variations of organic density. The result also indicates that, to fill the gap between $\kappa$chem and $\kappa$gf observed at noontime, the effective density of BC should be extremely high due to the decreased sensitivity of $\kappa$chem to BC density with the aging of BC. In this case, the assumed density of BC is 1.7 g cm-3, which reflects a very compacted and void free structure of the BC particles. The current applied value represents an upper limit for the effective density of ambient BC particles according to previous observations at a site near urban Beijing (Zhang et al., 2015), which suggested the aged BC is generally with effective density of 1.2 g cm-3. Using this ambient observed density would lead to further underestimation in $\kappa$chem. Our results exhibit the increase of the density of BC and organics cannot explain the disparity between $\kappa$chem and $\kappa$gf observed around noontime in summer. This just, on the other hand, verifies the photochemical aging/coating effect on the aerosols hygroscopisity. In addition, the coexisting hygroscopic and hydrophobic species may have a strong influence on the phase state of particles, also likely affecting chemical interactions between inorganic and organic compounds as well as the overall hygroscopicity of mixed particles (Peng et al., 2016). Further investigations are needed to verify this. Our study suggest that, to accurately parameterize the effect of BC aging on particles hygroscopisty, future investigations need to measure the effective density and morphology of ambient BC, in particularity in those regions with complex local sources... ..."

(b) According to the author's argument, aerosols both at noon and at midnight have core-shell structure, but why the $\kappa$chem/kgf is quite distinct? More detailed interpretation and discussion are necessary.

Re: At noontime, the rapid photochemical aging of BC particles leads to the core-shell structure in which certain secondary aerosol generated from photochemical reactions is thickly coated on the surface of BC. However, the condensation effect during nighttime is less significant (indicated by the smaller disparity between $\kappa$chem and $\kappa$gf) than the coating effect caused by aerosols photochemical aging at noontime, due to thinner coating layer formed on the pre-exist particles during nighttime or other factors influencing the particles hygroscopisity. We have included a statement in the revised manuscript (see lines 367-373) as follows, "...We propose the increased underestimation during polluted conditions is likely due to enhanced condensation of secondary hygroscopic compounds (e.g. nitrate, sulfate) on pre-existing aerosols at lower temperature and higher relative humidity at nighttime (Wu et al., 2008; Wang et al., 2016; An et al., 2019). However, such condensation effect during nighttime is less significant (indicated by the smaller disparity between $\kappa$chem and $\kappa$gf) than the coating effect caused by aerosols photochemical aging at noontime, likely due to thinner coating layer formed on the pre-exist particles during nighttime or other factors influencing the particles hygroscopisity...."

(2) L259, "Since a size-resolved BC mass concentration measurement was not available during the campaign, we use the bulk mass fraction of BC particles measured by the AE33 combining with size-resolved BC distribution in Beijing reported by Liu et al. (2018) to estimate $\kappa$chem." As far as I know, the instrument to measure the size distribution of BC in Liu et al. (2018) is a SP2, which gives the BC core diameter. It is necessary to explain how to convert this size distribution of BC core to the size distribution of ambient aerosols.

Re: We have provided a statement in the revised version as following (also see lines 260 -263), "....During the calculation, the BC core diameter measured by SP2 has been converted to the diameter of coated BC particles by multiplying factors of 1.4 and 2.6 under clean (with bulk BC mass concentrations <2 $\mu$g m-3) and polluted (with bulk BC mass concentrations >2 $\mu$g m-3) conditions respectively (Liu et al., 2018). ...."

(3) L227 and fig. 3. "the concentration of the hydrophilic mode increased quickly around noontime and in the early afternoon (12:00-16:00)", which is explained by a transformation of the particles from externally to internally mixing state. However, I

have different opinion. From Fig. 3a, it is evident that 40 nm particles after 12:00 were dominated by new particle formation (NPF). Therefore, the decrease of hydrophobic mode could be attribute to the extremely large amount of hydrophilic particles from NPF overwhelmed all other particles.

Re: Thanks a lot for the comments. We have revised and included an explanation of "In addition, it is evident that 40 nm particles after 12:00 were dominated by NPF (Fig. 3). Therefore, the increase of hydrophobic mode particles suggests that a large amount of hydrophilic particles are generated from NPF." in the revised manuscript (see lines 220-222).

(4) It will be better if the authors can discuss more on the similarities and differences of the hygroscopicity calculation at different sites.

Re: We have provided more details on clarify how we derive and calculate the particles hygroscopisity at different sites (lines 179-189) as follows, "…In addition, we also compare the results from the field campaigns with those from other two sites, Xingtai (XT: $37.18°$ NïijŇ$114.37°$ E), and Xinzhou (XZ: $38.24°$ NïijŇ$112.43°$ E), in North China Plain (Fig. 1). At XZ site, we use the hygroscopic parameter (defined as $\kappa$CCNc) from size-resolved CCN measurements (Zhang et al., 2014, 2016) for comparison. More detailed descriptions of the method to retrieve $\kappa$CCNc can be found in (Petters and Kreidenweis (2007). Both of the $\kappa$gf and $\kappa$CCNc are derived based on $\kappa$ -KoÌĹhler Theory (Petters and Kreidenweis, 2007). But, different from the $\kappa$gf measured by the HTDMA system which is operated at RH of 90%, the $\kappa$CCNc is derived by measuring aerosols CCN activity under the condition of supersaturations with relative humidity of >100%. Previous studies from filed measurements and laboratory experiments showed that the $\kappa$CCNc is generally slight larger or smaller than $\kappa$gf, but they are basically comparable and can well represent an overall aerosols hygroscopisity (e.g. Carrico et al., 2008; Wex et al., 2009; Good et al., 2010; Irwin et al., 2010; Cerully et al., 2011; Wu et al., 2013; Zhang et al., 2017)."

(5) There have been several studies revealing the uncertainty of calculating hygroscopicity using the mixing rule, but few can provide proper solution. Is it possible for the authors to propose parameterized modification on the $\kappa$chem to reduce the uncertainty? If so, this paper will be enormously improved and will be far distinct from other studies. For example, should we use lower BC density value during the rush hours?

Re: This is a good point. We have made a sensitivity test to examine the effect of density of BC on calculated $\kappa$chem , and included statements and discussions about this in the revised version (lines 319-346, Fig. 8) as follows (Fig. R4), "...We suppose that the large disparity between $\kappa$chem and $\kappa$gf is due to temporal variations in actual density of BC and organics caused by the particles aging and local sources. The externally-mixed BC particles are with fractal structure and chain-like aggregates and have been reported with effective density of 0.25-0.45 g cm-3(McMurry et al., 2002), While the BC particles in the calculation is assumed as void free with effective density of 1.7 g cm-3. Such inappropriate assumption would lead to an underestimation of BC volume fraction and thus the overestimation in $\kappa$chem during the traffic rush hour and cooking time when BC particles are mostly freshly emitted with uncompacted structure. In addition, the significant increase in volume fraction of POA during the late afternoon would result in a lower density of organics, which is expected to be smaller than the assumed one (1.2 g cm-3) in the calculation. A sensitivity test has been done to examine the effect of density of BC and organics on calculated $\kappa$chem (Fig. 7). The result shows that the $\kappa$chem value reduces by 16-33% when applying the BC effective density of 0.25-0.45 g cm-3. This basically explains the disparity during the traffic rush hour. However, the changes in $\kappa$chem are within $\pm4\%$ when changing the organic density from 1.0 (typical for POA) to 1.4 (typical for SOA) g cm-3, suggesting insensitivity of $\kappa$chem to variations of organic density. The result also indicates that, to fill the gap between $\kappa$chem and $\kappa$gf observed at noontime, the effective density of BC should be extremely high due to the decreased sensitivity of $\kappa$chem to BC density with the aging of BC. In this case, the assumed density of BC is 1.7 g cm-3, which reflects a very compacted and void free structure of the BC particles. The current applied value represents an upper

limit for the effective density of ambient BC particles according to previous observations at a site near urban Beijing (Zhang et al., 2015), which suggested the aged BC is generally with effective density of 1.2 g cm-3. Using this ambient observed density would lead to further underestimation in $\kappa$chem. Our results exhibit the increase of the density of BC and organics cannot explain the disparity between $\kappa$chem and $\kappa$gf observed around noontime in summer. This just, on the other hand, verifies the photochemical aging/coating effect on the aerosols hygroscopisity. In addition, the coexisting hygroscopic and hydrophobic species may have a strong influence on the phase state of particles, also likely affecting chemical interactions between inorganic and organic compounds as well as the overall hygroscopicity of mixed particles (Peng et al., 2016). Further investigations are needed to verify this. Our study suggest that, to accurately parameterize the effect of BC aging on particles hygroscopisty, future investigations need to measure the effective density and morphology of ambient BC, in particularity in those regions with complex local sources... ..."

(6) For several times, the current manuscript cited Zhang et al. (2017), which is one of the previous studies done by the same group on the same topic. Therefore, it is appropriate to make a clear statement of the unresolved issues in the previous paper or what improvement has been made to this study so that the reader can easily understand the novelty of this paper.

Re: We have included the following statement in the revised version (also see lines 77-82), "...In the atmosphere, the $\kappa$, which is related to the particle mixing state diversity, varies largely across the size range of ambient fine particles (Rose et al., 2010). Previous study only compared the measured$\kappa$to that calculated based on bulk chemical composition (Zhang et al., 2017). Using size-resolved, not bulk, chemical composition measurements in different seasons is expected to provide more comprehensive understanding and insights of how the aerosols mixing state influence on their hygroscopisity, motivating our analysis that employs size-resolved chemical composition measured by an HR-ToF-AMS in this study."

Other minor comments: (1) fig. 2 is not reader-friendly. Please work out some way to make the information more clear. Re: Revised. (2) fig.3. There are totally 12 sub-figures here. Please consider naming each sub-figures rather than the current way (which is not clearly demonstrated). Re: Revised. (3) L150 and L160, the full term and the abbreviations of probability density functions (PDF)should be provided the first time in the text. Re: Revised. (4) Fig. 5, L266, should be "slopes of linear fits and correlation coefficients". Re: Revised.

Please also note the supplement to this comment:
https://www.atmos-chem-phys-discuss.net/acp-2019-583/acp-2019-583-AC2-supplement.pdf
* * *

---

## Author Response (AR2)

**RESPONSES TO REVIEWERS' COMMENTS**

Dear ACP Editorial Board,

We are submitting our revised paper entitled "Contrasting ambient fine particles hygroscopicity derived by HTDMA and HR-AMS measurements between summer and winter in urban Beijing." We are grateful to the two reviewers for their insightful and constructive comments and have revised our paper accordingly to account for the reviewers' recommendations. Below please find our detailed point-by-point responses (in blue) to the reviewers' comments (in black) to the manuscript. We believe that we have satisfactorily addressed all criticisms from the two reviewers.

Thank you for your attention to this matter.

Sincerely, Fang Zhang on behalf of all authors

**Anonymous Referee #1**

Zhang et al. present a study comparing water uptake and predicted water uptake of aerosols in Beijing and North China. The authors have addressed reviewer comments and present a revised manuscript that is considerably clearer to read. I recommend publication once the following comments are addressed:

(1) Novelty
  The novelty of the study could be improved. Mei et al. (2013), Zhang et al. (2016) and Zhang et al. (2017) have compared kappa derived from composition measurements to kappa measured by co-located instruments. Mei et al. (2013) derive an empirical relationship between f44 and kappa_org based on their observations. This relationship may not apply to dissimilar field sites. The correlation between f44 or O:C ratio and kappa is not always linear. The correlation between oxidation and kappa is strong but nevertheless an empirical relationship resulting from underlying molecular composition. Zhang et al. (2016), Zhang et al. (2017), and the current study discuss over- and under-prediction of kappa for their measurements. I think it is generally acknowledged that different emission profiles result in different chemistry – I recommend discussing these differences, and perhaps de-emphasizing the poor representation of the measurement by published empirical models. This model/measurement disagreement has already been discussed (Zhang et al., 2016, Zhang et al., 2017).

Re: we appreciate the comments. We have revised the corresponding discussions in the paper as follows (or lines 377-392),

"…The uncertainty in calculation of $\kappa_{chem}$ may be also related to the uncertainty caused by composition of organics that vary widely over a range of diverse constitutes of SOA (Suda et al., 2012). The lower $\kappa_{chem}$ indicates that the $\kappa$ of secondary organic aerosols formed through the strong photochemical oxidation processes in summer of urban Beijing are likely underestimated. In this study, the mean $\kappa$ value of organics derived from the $f_{44}$ parametrized equation is $0.20\pm$ 0.02, ranging from 0.17 to 0.23 during 09:00-17:00. While the organic aerosols, especially for particles in accumulated mode, may be more hygrophilic with much larger $\kappa$, i.e. >0.2 due to large formation of highly-oxidized OA. One can easily get that increasing the $\kappa$ of organic aerosols from 0.2 to 0.3 can explain about 11-13% underestimation of $\kappa_{chem}$, but representing an upper limit of the impact of hygroscopisty of organic aerosols on the calculation. This is because that the $\kappa$ value of 0.3 corresponds to the maximum possible for ambient organic aerosols. Additionally, the $f_{44}$ parametrized equation tends to overestimate the $\kappa$ according to Fröhlich et al. (2015), which should yield a larger $\kappa_{chem}$. Finally, the coexisting hygroscopic and hydrophobic species may have a strong influence on the phase state of particles, also likely affecting chemical interactions between inorganic and organic compounds as well as the overall hygroscopicity of mixed particles (Peng et al., 2016). Overall, The lower $\kappa_{chem}$ caused by the photochemical aging effect is likely resulted from multiple impacts of inappropriate application of density and hygroscopic parameter of organic aerosols in the calculation, as well as the influences from chemical interaction between organic and inorganic compounds on the overall hygroscopicity of mixed particles. This topic warrants further investigations."

(2) Coating effect

I think that the coating effect on hygroscopicity should be discussed with appropriate caveats. I find it unlikely that a coating of organic material on an inorganic core limits uptake of water as implied. Studies have shown that water diffusion into viscous organics is fast. These rates are published and will show that for nanoscale aerosol, it is unlikely that condensing SOA prevents water uptake by an inorganic core. Various other quantities go into kappa_chem, including e.g. density as the other reviewer has mentioned and also composition (and hygroscopicity) of the organic. These are good candidates for the underlying mechanisms causing poor agreement between model and measurement.

Re: Thanks a lot for the comments. We agree with that to address the coating effect with appropriate caveats. We have revised the corresponding discussions and statements in the paper (see revised Abstract, Section 3.4 and Conclusions). Specifically, previous statements about coating effect have been presented as "aging effect". We have provided revised statements of following to explain the lower $\kappa_{chem}$ around noontime in summer,

"The lower $\kappa_{chem}$ caused by the photochemical aging effect is likely resulted from multiple impacts of inappropriate application of density and hygroscopic parameter of organic aerosols in the calculation, as well as the unknown influences from chemical interaction between organic and inorganic compounds on the overall hygroscopicity of mixed particles."

(3) Technical corrections

The manuscript would benefit from a careful reading to correct technical errors. Two examples are included here, but there are many more.

Re: Careful corrections of technical errors throughout the manuscript have been done.

Lines 27-28: This sentence needs reworking. "the hypothesis" is mentioned before it is defined.

Re: Revised.

Line 29: on average

Re: Revised.

(4) References:

Mei, F., Setyan, A., Zhang, Q., and Wang, J.: CCN activity of organic aerosols observed downwind of urban emissions during CARES, Atmos. Chem. Phys., 13, 12155–12169, https://doi.org/10.5194/acp-13-12155-2013, 2013.

Zhang, F., Wang, Y., Peng, J., Ren, J., Collins, D., Zhang, R.,...Li, Z. (2017). Uncertainty in predicting CCN activity of aged and primary aerosols. Journal of Geophysical Research: Atmospheres, 122. https://doi.org/10.1002/2017JD027058

Zhang, F., Li, Z., Li, Y., Sun, Y., Wang, Z., Li, P., Sun, L., Wang, P., Cribb, M., Zhao, C., Fan, T., Yang, X., and Wang, Q.: Impacts of organic aerosols and its oxidation level on CCN activity from measurement at a suburban site in China, Atmos. Chem. Phys., 16, 5413–5425, https://doi.org/10.5194/acp-16-5413-2016, 2016.

**Anonymous Referee #2**

This revised manuscript has addressed all my previous concerns. Meanwhile, the further discovery that an inappropriate use of BC density may cause significant overestimation of the hygroscopic parameter, is really intriguing, making the current version much better in terms of scientific novelty. I suggest the publication of this manuscript in ACP after a minor revision. The disparity between $\kappa_{chem}$ and $\kappa_{gf}$ during rush hour could also be attributed to the bias from the HTDMA measurements. The HTDMA may overestimate the $D_{dry}$ for the external mixed fractal BC particles, as BC-containing particles may shrink when humidified, leading to underestimate the hygroscopic growth factor.

Re: We have included a statement to address the uncertainty caused by the HTDMA techniques in the revised version (see lines 273-275).

The language and expression need to be improved. Some examples are listed as follows.

Re: Careful corrections of technical errors throughout the manuscript have been done.

Line 57, the sentence could be revised as "In a heavily polluted atmosphere with varied aerosol sources and sinks as well as complex physical and chemical processes, the mixing state and its impact on aerosols hygroscopicity is more complicated."

Re: Revised.

Line 61, this is not a complete sentence.

Re: Corrected and Revised.

Line 69, "during the periods dominated by primary emissions"

Re: Revised.

Line 76, "has been lacking" is really a weird expression.

Re: Revised as "A comprehensive investigation on the causes and magnitude of the effect is with great significance to parameterize the effect of atmospheric processes/emissions of aerosols on particles hygroscopicity in models."

Line 83, this sentence is weird, try revising it.

Re: Revised.

Line 289, change "which" to "it".

Re: Corrected.

Line 302, delete "and"

Re: Deleted.

[revised manuscript text omitted]

---

## Author Response (AR3)

**Author's response**

Editor's comments to the Author:

Please consider making the data more accessible, as per ACP's data policy:

https://www.atmospheric-chemistry-and-physics.net/about/data_policy.html

Re: A website with DOI number has been provided in the "Data availability" of the paper, as follows,

*"Data availability.* All data used in the study are available on http://www.geodoi.ac.cn/WebCn/doi.aspx?Id=1356 (doi:10.3974/geodb.2019.06.11.V1) or from the corresponding author upon request (fang.zhang@bnu.edu.cn)."

Thank you for your attention to this matter.

Sincerely,
Fang Zhang on behalf of all authors